# Poly(1,3-Propylene Glycol Citrate) as a Plasticizer for Toughness Enhancement of Poly-L-Lactic Acid

**DOI:** 10.3390/polym15102334

**Published:** 2023-05-17

**Authors:** Dengbang Jiang, Junchao Chen, Minna Ma, Xiushuang Song, Huaying A, Jingmei Lu, Conglie Zi, Wan Zhao, Yaozhong Lan, Mingwei Yuan

**Affiliations:** 1Green Preparation Technology of Biobased Materials National &Local Joint Engineering Research Center, Yunnan Minzu University, Kunming 650500, China; 041814@ymu.edu.cn (D.J.); cjc9931@163.com (J.C.);; 2School of Ecology and Environmental Science, Yunnan University, Kunming 650091, China; ynulanyaozhong@163.com

**Keywords:** polylactic acid, citric acid, poly(1,3-propylene glycol), toughening agent

## Abstract

Despite the unique features of poly-L-lactic acid (PLLA), its mechanical properties, such as the elongation at break, need improvement to broaden its application scope. Herein, poly(1,3-propylene glycol citrate) (PO3GCA) was synthesized via a one-step reaction and evaluated as a plasticizer for PLLA films. Thin-film characterization of PLLA/PO3GCA films prepared via solution casting revealed that PO3GCA shows good compatibility with PLLA. The addition of PO3GCA slightly improves the thermal stability and enhances the toughness of PLLA films. In particular, the elongation at break of the PLLA/PO3GCA films with PO3GCA mass contents of 5%, 10%, 15%, and 20% increases to 172%, 209%, 230%, and 218%, respectively. Therefore, PO3GCA is promising as a plasticizer for PLLA.

## 1. Introduction

Polylactic acid (PLA) is a biodegradable and eco-friendly polymer that has garnered significant attention in recent years due to its numerous advantages. PLA is derived from renewable plant resources, such as corn, potato, and jackfruit, through a fermentation process that produces high-purity lactic acid. The lactic acid is then used to synthesize PLA of the desired molecular weight through chemical processes [1,2,3].

PLA is biocompatible and is widely used in the medical field, including as implant materials, drug delivery systems, and tissue engineering scaffolds [4]. It is also used in commodity packaging, where its biodegradability and low environmental impact make it an attractive alternative to traditional plastics [5]. Additionally, PLA has been explored for use in textiles due to its biodegradability and potential to reduce the environmental impact of textile production [6].

Despite its numerous advantages, PLA has limitations that have hindered its widespread application. One of the major drawbacks of PLA is its high glass transition temperature (Tg) which ranges from 55 °C to 65 °C, making it brittle and fragile at room temperature. Additionally, the elongation at the break of PLA is limited to only ~5%, which further limits its practical use [7]. Efforts have been made to improve the mechanical properties of PLA through various methods, including blending it with other polymers, copolymerization, and the addition of nanoparticles [7,8,9].

To improve the properties and performance of Polylactic acid (PLA), two commonly used methods include the copolymerization of lactic acid with other polymer monomers, such as trimethylene carbonate and caprolactone [8,9], and the preparation of physical blends of PLA with different polymers, small molecular compounds, or even inorganic materials. These approaches have shown great potential in enhancing the thermal stability, mechanical strength, and elongation at the break of PLA. Furthermore, they have opened up new avenues for the diversified application of PLA in various fields, including automotive, construction, and electronics.

However, there are some challenges associated with these methods that need to be addressed. For instance, the compatibility of the blended systems needs to be optimized to achieve the desired properties of PLA-based materials. The processing conditions, such as temperature, pressure, and blending time, also need to be carefully controlled to obtain the desired properties of PLA-based materials [10,11,12].

Despite these challenges, research efforts have been ongoing to overcome these limitations and optimize the properties of PLA-based materials. A better understanding of the fundamental properties of PLA and the mechanisms underlying the modification of its properties will enable researchers to design new strategies for enhancing the performance of PLA-based materials. Moreover, the development of sustainable and cost-effective methods for the production of PLA will further promote the widespread application of this eco-friendly polymer.

In the process of blending modification, a plasticizer is commonly used to enhance the properties of Polylactic acid (PLA). The ideal plasticizer should exhibit non-toxicity, excellent compatibility with PLA, and high thermal stability [11,12,13]. Citrate is a plasticizer that meets these requirements, and various citrate derivatives, such as triethyl citrate (TEC), acetyl tributyl citrate (ATBC), tributyl citrate (TBC), and polyethylene glycol citrate, have been utilized as plasticizers for PLA [14]. For example, Ljungberg N et al. toughened PLA by incorporating TBC and ethyl triacetate (TAC). Although the addition of TBC did not significantly affect the crystallinity of PLA, it increased the elongation at the break of PLA/TBC blends up to 350%, considerably improving the toughness of PLA [13]. Mounira Maiza prepared and characterized PLA/TEC and PLA/ATBC blends using TEC and ATBC as plasticizers for PLA, respectively. TEC and ATBC effectively reduced the Tg and enhanced the crystallinity of the blends, while having no substantial effect on their transparency [14]. Gui Zongyan demonstrated that the incorporation of polyethylene glycol (PEG) and carboxylic acid copolymers could effectively enhance the toughness of PLA [12]. Further research is required to optimize the processing conditions and to investigate the compatibility of the blended systems, to achieve the desired properties of PLA-based materials. Poly(1,3-propylene glycol) (PO3G) is a hydrophilic polymer synthesized using 1,3-propylene glycol via an acid-catalyzed dehydration process. Although its structure is highly similar to that of PEG, PO3G has better compliance, mechanical properties, and thermal stability, which stem from its longer molecular chain [15,16]. Therefore, we envisioned that PO3G or its combination with other carboxylic acids could serve as a plasticizer for PLA.

This study utilized a novel approach to improve the properties of Polylactic-L-acid (PLLA) by physically blending it with poly(1,3-propylene glycol citrate) (PO3GCA), a copolymer obtained from biologically derived citric acid (CA) and PO3G. The aim of this study was to investigate the plasticizing effect of PO3GCA on PLLA. This approach is unique, as it involves the use of a copolymer obtained from natural sources to enhance the properties of PLLA.

The results of this study have significant implications for the development of biodegradable and eco-friendly plasticizers for PLLA, as well as for the sustainable utilization of renewable resources. By utilizing natural sources to synthesize copolymers, it is possible to reduce the environmental impact of plasticizers and promote the use of biodegradable materials. Moreover, the use of PO3GCA as a plasticizer for PLLA may offer enhanced mechanical properties, thermal stability, and elongation at break, making it a promising alternative to traditional plasticizers.

Overall, this study highlights the importance of developing sustainable and eco-friendly materials for the plastic industry. The use of renewable resources and biodegradable materials can significantly reduce the environmental impact of plastic production and promote the development of a more sustainable future.

## 2. Materials and Methods

### 2.1. Raw Materials

PLLA (4032D, Nature Works, Minnetonka, MN, USA) [17], PO3G (Mn = 2300, DuPont Company, Wilmington, DE, USA), Citric acid monohydrate (McLean Biochemical Technology Co., Ltd., Shanghai, China, 99.5%), 1,4-dioxane (Adamas Reagent Company, Shanghai, China), molybdenum trioxide (Adamas Reagent Company, Shanghai, China, 99.9%), and methylene chloride (DCM, Damao Chemical Reagent Factory, Tianjin, China) were used as received.

### 2.2. Experimental Equipment

The following instruments were used for the experiments: a heat-collection constant temperature heating magnetic agitator (DF-101S, Yuhua Instrument Co., Ltd., Gongyi, China), an electric blast drying oven (GZX-9240MBE, Medical Equipment Factory of Boxun Industrial Co., Ltd., Shanghai, China), an electronic balance (FA2004, Yueping Scientific Instrument Co., Ltd., Shanghai, China), a nuclear magnetic resonance (NMR) spectrometer (400 MHz, BrukerAvance-II, Bruker, Bremen, Germany), a Fourier transform infrared (FTIR) spectrometer (Nicolet IS10, Thermo Fischer Scientific, Waltham, MA, USA), a scanning electron microscope(EvoMA10, ZEISS, Oerkochen, Germany),a microcomputer-controlled universal tensile testing machine (CMT4104, Chuangcheng Zhijia Technology Co., Ltd., Beijing, China), a thermogravimetric analyzer (STA449F3, NETZSCH, Selb, Germany), a differential scanning calorimeter (2414Polyma, NETZSCH, Selb, Germany).

### 2.3. Raw Materials

#### 2.3.1. Synthesis of PO3GCA

In this study, the PO3GCA copolymer was synthesized using a direct synthesis method [12,18,19,20,21,22]. Specifically, PO3G (300 g) and CA (21 g) were added to a 500 mL three-necked flask in a 1.3:1 molar ratio, with 0.5% (1.6 g) molybdenum trioxide used as the catalyst. After purging the reaction system with nitrogen gas three times, the system was evacuated to a pressure of 100 Pa using an oil pump. The system was gradually heated to 150 °C and held for 30 min, followed by heating at 170 °C for 8 h. Mechanical stirring was employed throughout the entire reaction process. After natural cooling to room temperature, the product was dissolved in dichloromethane (DCM) and stirred magnetically for 1 h. The solution was then left to stand for 10 h until most of the molybdenum trioxide precipitated at the bottom of the flask. The liquid above the precipitate was filtered and collected, and the filtrate was filtered 2–3 times to remove all molybdenum trioxide. The resulting filtrate was placed in the flask and distilled at 50 °C until no more vapor was produced. The second distillation was carried out at 110 °C until no more vapor was produced, yielding approximately 290 g of a light yellow to yellow, viscous liquid, with a yield of approximately 90.3%. This light yellow to yellow, viscous liquid is PO3GCA.

Figure 1 shows the reaction equation of PO3GCA synthesized from poly (1, 3-propylene glycol) and citric acid.

#### 2.3.2. Preparation of PLLA/PO3GCA Films

The study involved the preparation of PLLA/PO3GCA composite films with a total mass concentration of 20%. The proportion of PO3GCA to solute ranged from 0% to 20%, with increments of 5% (The solvent used for the solution was 1,4-dioxane). The solution was stirred until it reached a temperature of 80 °C, followed by ultrasonic defoaming. The solution was then scraped onto a clean glass plate using a film scraper (750 μm) and allowed to dry after solvent volatilization. The resulting composite film samples had a uniform thickness of 0.07–0.08 mm and were naturally dried at room temperature for 48 h before being collected and stored for further use. The samples were labeled PLLA, PO3GCA5, PO3GCA10, PO3GCA15, and PO3GCA20, according to the concentration of PO3GCA.

### 2.4. Characterization of PO3GCA and the PLLA/PO3GCA Films

#### 2.4.1. ^1^H NMR Spectroscopy Characterization of PO3GCA

The samples for ^1^H NMR spectroscopy analysis were prepared by placing 15 mg of PO3GCA in an NMR tube and dissolving it using 480 μL of Deuterium chloroform. Tetramethylsilane was used as the internal standard.

#### 2.4.2. FTIR Spectroscopy Characterization of PO3GCA

KBr pellets were prepared by mixing and grinding dried PO3GCA powder and KBr in a mass ratio of 2:100, followed by pressing with a tablet press. The FTIR spectra were recorded over a range of 400–4000 cm^−1^.

#### 2.4.3. Tensile Test of the PLLA/PO3GCA Films

The PLLA/PO3GCA films were cut into smooth and nonwound samples with a size of 20 mm × 60 mm. At ambient temperature (25 °C), the initial length of the sample was 50 mm, the tensile rate was 10 mm/min, and the relative humidity was 60%. The sample was fixed on the fixture of the testing machine. The clamping length of the sample film was 50 mm, the force measurement accuracy was 0.01 cN, and the elongation accuracy was 0.01 mm. Stress–strain curves were obtained by testing each sample on a microcomputer-controlled electronic universal testing machine five times. The elongation at the break of each PLLA/PO3GCA film was determined using Equation (1).
(1)$$\mathrm{Elongation}\;\mathrm{at}\;\mathrm{break}=\frac{\mathrm{Fracture}\;\mathrm{displacement}\;\mathrm{value}(\mathrm{mm})}{\mathrm{The}\;\mathrm{initial}\;\mathrm{length}\;\mathrm{of}\;\mathrm{the}\;\mathrm{membrane}(\mathrm{mm})}\times 100\%\;$$

#### 2.4.4. Differential Scanning Calorimetry (DSC) Characterization of the PLLA/PO3GCA Films

The thermal properties of PLLA/PO3GCA thin films were analyzed using differential scanning calorimetry (DSC). Film samples weighing approximately 5–8 mg were placed in an aluminum crucible and heated from −40 °C to 200° C at a rate of 10 °C/min to obtain the DSC curve of the film samples [23].

The crystallinity of the composite films was calculated using Equation (2) [24,25], where *X*_c_ denotes crystallinity, Δ*H*_m_ denotes the enthalpy of melting, Δ*H*_cc_ denotes the enthalpy of cold crystallization, Δ*H_mPLLA_* denotes the standard enthalpy of melting of PLLA (93.6 J·g), and *ω_PLLA_* represents the mass fraction of PLLA in the composite film.
(2)$$X_{c}=\frac{\Delta H_{m}-\Delta H_{\mathrm{cc}}}{\Delta H_{mPLLA}{}_{{}_{}}\times \omega_{PLLA}{}_{}}\times 100\%$$

#### 2.4.5. Field-Emission Scanning Electron Microscopy (SEM) Characterization of the PLLA/PO3GCA Films

The morphology and fracture surface of the PLLA/PO3GCA films were examined using field-emission scanning electron microscopy (SEM). For surface morphology analysis, fully dried films were affixed to the sample table using conductive carbon glue and coated with a thin layer of gold for 30 s under vacuum conditions to enhance their conductivity. The samples were then observed at various magnifications at an accelerating voltage of 2 kV following vacuum extraction, and the corresponding images were captured for further analysis. For fracture surface analysis, the cross-sections of sample bars pulled at a rate of 10 mm/min on a universal tensile testing machine were used for SEM testing. Prior to testing, the surface of the samples was coated with a layer of conductive metal to enhance their conductivity, and then the samples were fractured to expose their internal structure. The SEM imaging process involved several steps, including surface treatment of the samples, vacuum extraction to remove air and moisture, and observation and capture of images at various magnifications in the SEM. This rigorous sample preparation and imaging protocol ensured the high-resolution imaging of the PLLA/PO3GCA films, allowing for a detailed examination of their microstructure, surface characteristics, and fracture behavior.

#### 2.4.6. Thermogravimetric Analysis (TGA) of the PLLA/PO3GCA Films

The thermal stability of the PLLA/PO3GCA films was evaluated via TGA from room temperature (23 °C) to 600 °C at a heating rate of 10 °C/min under flowing nitrogen. The differential thermogravimetric (DTG) curves were obtained by differentiating the TGA curves.

## 3. Results and Discussion

### 3.1. Characterization of PO3GCA

Figure 2 shows the FTIR spectra of PO3G, PO3GCA, and CA. In the graph of PO3G, the peaks at 3500 and 1108 cm are the stretching vibrations of the terminal OH and ether bond C-O. The ester absorption peak at 1737 cm in the graph of PO3GCA shifted to the right compared to the carboxyl absorption peak at 1728 cm in the graph of CA, indicating that an esterification reaction occurred. The OH absorption peak near 3500 cm is significantly weaker than the OH absorption peak in the PO3G spectrum, which also indicates that OH participates in the esterification reaction.

Figure 3 shows the ^1^H NMR spectra of CA, PO3G, and PO3GCA. In the spectrum of CA, the two peaks at 2.7–2.7 ppm correspond to –CH_2_–(C=O)OH. In the spectrum of PO3G, the peaks at 3.55, 3.5, and 1.8 ppm can be attributed to the –CH_2_–OH–CH_2_–CH_2_–O, and –CH_2_–CH_2_– groups, respectively; the integral areas of these peaks were determined to be 1, 19.57, and 39.12, respectively. For PO3G, the average degree of polymerization was ~3 and Mn was 2260, which is close to the molecular weight of the raw material. The peak at 3.55 ppm in the spectrum of PO3G moves to 4.2 ppm in the spectrum of PO3GCA, indicating the formation of a–CH_2_–O(C=O)– group as a result of the reaction of the hydroxyl group of PO3G with CA.

### 3.2. Characterization of the PLLA/PO3GCA Films

#### 3.2.1. Structure and Thermal Properties

##### TGA

Thermal properties of PLLA/PO3GCA composite films and PO3GCA were analyzed using thermogravimetric (TG) and derivative thermogravimetric (DTG) curves, as shown in Figure 4 and Figure 5, respectively. The weight loss of PLLA and PLLA/PO3GCA composite membranes occurred in the temperature range of 75–140 °C, which was attributed to the evaporation of bonded water and residual solvents in the membranes. In contrast, the weight loss of PO3GCA was not evident in this temperature range. The PLLA thin film experienced a second weight loss at 295–390 °C, while the PLLA/PO3GCA composite thin film experienced a second weight loss at 300–430 °C. PO3GCA showed slight weight loss in the temperature range of 270–310 °C, which was partly due to the thermal degradation of low molecular weight PO3GCA. PO3GCA also showed a secondary weight loss at 310–450 °C. Both types of films exhibited 50% weight loss at 360 °C, while PO3GCA exhibited 50% weight loss at 380 °C. The maximum thermal decomposition rate of the film during the second weight loss was negatively correlated with the content of PO3GCA, as shown in Figure 5. With the increase of PO3GCA content, the maximum thermal decomposition rate decreased. These results indicate that the addition of PO3GCA can effectively improve the thermal stability of PLLA thin film. Overall, the TG and DTG analyses provided valuable insights into the thermal behavior of PLLA/PO3GCA composite films and highlighted the potential of PO3GCA as a thermal stabilizer for PLLA.

##### DSC Analysis

The PLLA/PO3GCA composite films with varying PO3GCA content and DSC curves of PO3GCA are shown in Figure 6. The glass transition temperature (Tg) of PO3GCA is 8.2 °C. The Tg, cold crystallization temperature (Tcc), and melting temperature (Tm) of the polylactic acid (PLLA) film are 62.3 °C, 123.3 °C, and 165.4 °C, respectively. As can be seen from Figure 6 and Table 1, with increasing PO3GCA content, the Tg of the PLLA/PO3GCA composite films gradually decreases compared to that of the pure PLLA film. This indicates that the addition of PO3GCA improves the mobility of the PLLA segments, possibly due to the good compatibility between PLLA and PO3GCA. Moreover, when the PO3GCA content increases from 10% to 15%, a second glass transition temperature appears at 8.7 °C, which is close to the Tg of PO3GCA, indicating that slight phase separation occurs between PO3GCA and PLLA. Additionally, the second Tg at 43.7 °C is higher than that of PO3GCA10 at 37.7 °C. When the PO3GCA content increases to 20%, the phase separation becomes more severe, and the Tg increases to 45.3 °C.

Furthermore, it is worth noting that when the PO3GCA content is less than 10%, the peak cold crystallization temperature of the composite film decreases with the increasing PO3GCA content. This indicates that the addition of PO3GCA effectively promotes the cold crystallization of PLLA, which is likely due to the nucleation effect of PO3GCA. Specifically, PO3GCA provides more nucleation sites for the formation of PLLA crystals, thereby enhancing the crystallization process. Additionally, as the PO3GCA content increases, the melting peak of the composite film slightly broadens and shifts to a lower temperature, indicating that the addition of PO3GCA affects the melting behavior of PLLA. Overall, the DSC analysis provides valuable insights into the thermal properties and compatibility of the PLLA/PO3GCA composite films.

As depicted in Table 1, it is evident that when the PO3GCA content is lower than 10%, the crystallinity of the composite film is positively correlated with the PO3GCA content, and the crystallinity of the film is significantly improved. This suggests that the addition of PO3GCA can effectively enhance the crystallization property of PLLA under suitable conditions. However, when the PO3GCA content exceeds 15%, the crystallinity of the composite film begins to decline. This is because the excessive addition of PO3GCA results in the separation of PO3GCA from PLLA, which is not conducive to the crystallization of PLLA. Therefore, it is crucial to maintain an appropriate PO3GCA content to achieve the desired crystallization improvement effect. In addition, it should be noted that the improvement in crystallinity is also closely related to the compatibility between PO3GCA and PLLA. When the compatibility is good, the nucleation effect of PO3GCA on PLLA crystallization is enhanced, resulting in higher crystallinity. Overall, the crystallinity analysis provides valuable insights into the crystallization behavior of the PLLA/PO3GCA composite films and highlights the importance of maintaining an appropriate PO3GCA content and good compatibility with PLLA.

##### Field-Emission SEM 

Figure 6 and Figure 7 present the surface micromorphology and micromorphology of stretched sections of PLLA/PO3GCA films with varying PO3GCA contents. The images in Figure 7 demonstrate that as the PO3GCA content increased, the surface of the films changed from a uniform phase to a dispersed phase with a sea-island structure. Additionally, the distribution of PO3GCA in PLLA shifted from the interior to the surface, particularly when the PO3GCA content increased from 10% to 15%. These observations suggest that the incorporation of PO3GCA into PLLA can significantly alter the morphology of the composite films, which could have implications for their mechanical and functional properties.

The tensile fracture sections shown in Figure 8 revealed that stratification occurred in the PLLA/PO3GCA15 and PLLA/PO3GCA20 films, whereas this phenomenon was not observed in the PLLA/PO3GCA5 and PLLA/PO3GCA10 films. This finding indicates that PO3GCA may be included in the cracks of the PLLA chains, filling the gaps between PLLA layers. However, excess PO3GCA cannot be accommodated in these gaps, resulting in the stratification of the film and the formation of a sea-island structure of dispersed phases on the surface. Furthermore, scattered fine particles were observed on the surface of the PLLA/PO3GCA5 and PLLA/PO3GCA10 films, which are most likely PO3GCA macromolecules that cannot be accommodated into the gaps between the PLLA layers. Another possibility is that the PO3GCA plasticizer crystallizes in advance during the solvent volatilization process owing to its high molecular weight, promoting the aggregation of the surrounding plasticizer and resulting in scattered particles on the film surface.

The fracture surface of the PLLA/PO3GCA blend films was rough, with a distinct fibrous surface visible at a PO3GCA content of 15%, indicating a ductile fracture corresponding to a high elongation at break and high toughness of PLLA/PO3GCA15. As the PO3GCA content increased from 5% to 20%, burrs on the fracture surface became increasingly thin, indicating that PO3GCA may enhance the toughness of the PLLA film. However, the plasticizer overflowed between layers and accumulated on the surface as the PO3GCA content increased, thereby decreasing the tensile properties and toughness. These findings suggest that the optimal PO3GCA content for improving the mechanical properties of PLLA/PO3GCA films should be carefully considered.

#### 3.2.2. Mechanical Properties of the PLLA/PO3GCA Films

Figure 9 illustrates the stress-strain curves obtained from the mechanical testing of PLLA/PO3GCA films with varying compositions. PLLA films are inherently rigid and brittle, with an elastic modulus of 1528 MPa and an elongation at a break of less than 10%. As shown in Table 2, the addition of PO3GCA plasticizer had a significant impact on the elongation at the break of the films. Among the different hybrid systems tested, the PLLA/PO3GCA15 film exhibited the highest elongation at break, which increased by 220% compared to pure PLLA.

The stress-strain curves (Figure 9) indicate that the addition of PO3GCA resulted in plastic deformation of the PLLA/PO3GCA films due to changes in the interaction at the PLLA interface. The interfacial interaction between PO3GCA and the PLLA matrix facilitated the sliding of the interface of the hybrid matrix film, leading to a reduction in the tensile yield stress relative to the maximum force of the pure PLLA film. However, when the PO3GCA content reached 20%, the mechanical properties of the corresponding film decreased to varying degrees. This was attributed to excessive PO3GCA filling the gaps between the PLLA layers and aggregating on the surface of the film, causing the PLLA layer of the mixed films to slip easily and affecting the intralaminar structure of PLLA, ultimately leading to film fracture. As the elongation decreased, the elastic modulus, tensile yield stress, and maximum force also decreased.

In summary, the addition of PO3GCA significantly improved the mechanical properties of PLLA films, particularly in terms of elongation at break and toughness. However, excessive PO3GCA content can negatively impact the mechanical properties of PLLA/PO3GCA films. These findings have important implications for the use of PO3GCA as a plasticizer for PLLA in various applications, such as food packaging and medical devices.

The tensile toughness of the films was determined by integrating the stress-strain curves [26], and the results are presented in Table 2. The pure PLLA film exhibited a toughness of 1.47 MPa. However, upon the incorporation of PO3GCA, a substantial improvement in the toughness of the films was observed. Specifically, the toughness values for PLLA/PO3GCA5, PLLA/PO3GCA10, PLLA/PO3GCA15, and PLLA/PO3GCA20 were found to be 38.14, 44.89, 44.69, and 34.33 MPa, respectively. These values represent a remarkable increase of 2495%, 2953%, 2940%, and 2235% compared to that of pure PLLA. These findings indicate that PO3GCA is an excellent plasticizer for PLLA, and has the potential to significantly enhance the mechanical properties of PLLA-based materials.

## 4. Conclusions

In this study, poly(1,3-propylene glycol citrate) (PO3GCA) was synthesized from biologically derived citric acid (CA) and PO3G and was used as a toughening agent for Polylactic-L-acid (PLLA). The solution casting method was employed to prepare PLLA/PO3GCA films with varying PO3GCA contents while ensuring satisfactory compatibility between PO3GCA and PLLA.

The addition of PO3GCA to PLLA resulted in a slight increase in the thermal stability of the PLLA film and a significant improvement in its toughness. The elongation at the break of the composite film reached 230% when the PO3GCA content was 15%, compared to the pure PLLA film. These results suggest that PO3GCA may be a promising plasticizer for PLLA films, as it enhances the toughness of the film while maintaining its biodegradability and environmental-friendliness.

Overall, this study contributes to the development of sustainable and eco-friendly materials for the plastic industry. The use of biologically derived citric acid and poly(1,3-propylene glycol citrate) as toughening agents for PLLA can significantly enhance the properties of PLLA-based materials, while also reducing the environmental impact of plastic production. Further research is needed to investigate the long-term stability and biodegradability of PLLA/PO3GCA composite films, as well as to optimize the processing conditions and compatibility of the blended systems to achieve the desired properties of PLLA-based materials.

## Figures and Tables

**Figure 1 polymers-15-02334-f001:**
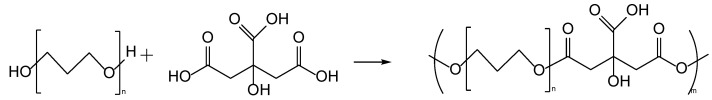
Reaction equation for the synthesis of PO3GCA from poly1,3-propylene glycol, and citric acid.

**Figure 2 polymers-15-02334-f002:**
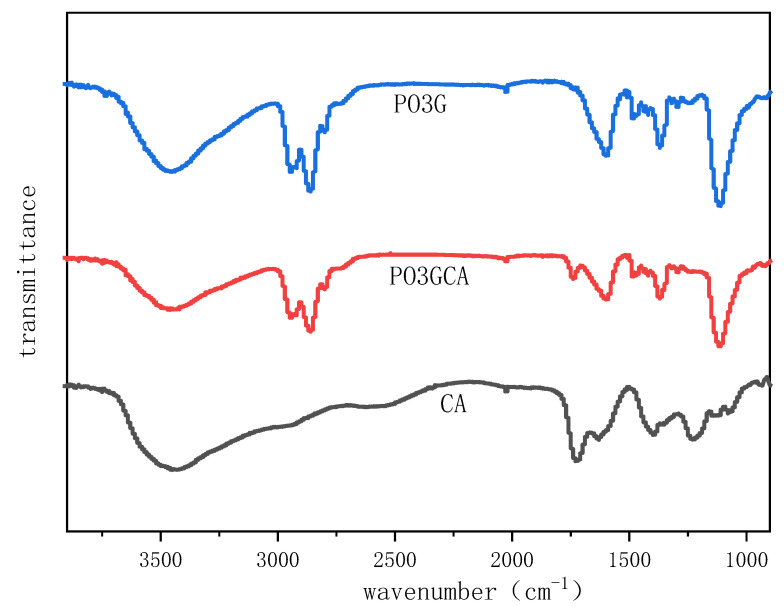
Fourier transform infrared (FTIR) spectra of poly(1,3-propylene glycol) (PO3G), poly(1,3-propylene glycol citrate) (PO3GCA), and citric acid (CA).

**Figure 3 polymers-15-02334-f003:**
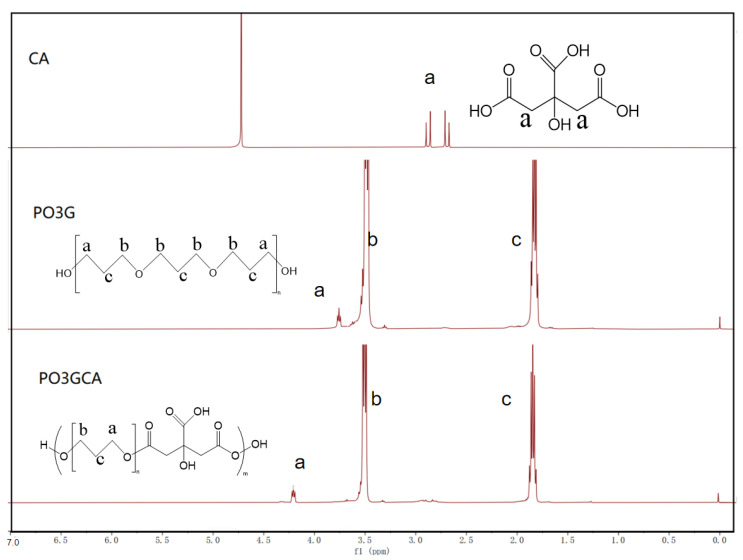
^1^H nuclear magnetic resonance (NMR) spectra of citric acid (CA), poly(1,3-propylene glycol) (PO3G), and poly(1,3-propylene glycol citrate) (PO3GCA).

**Figure 4 polymers-15-02334-f004:**
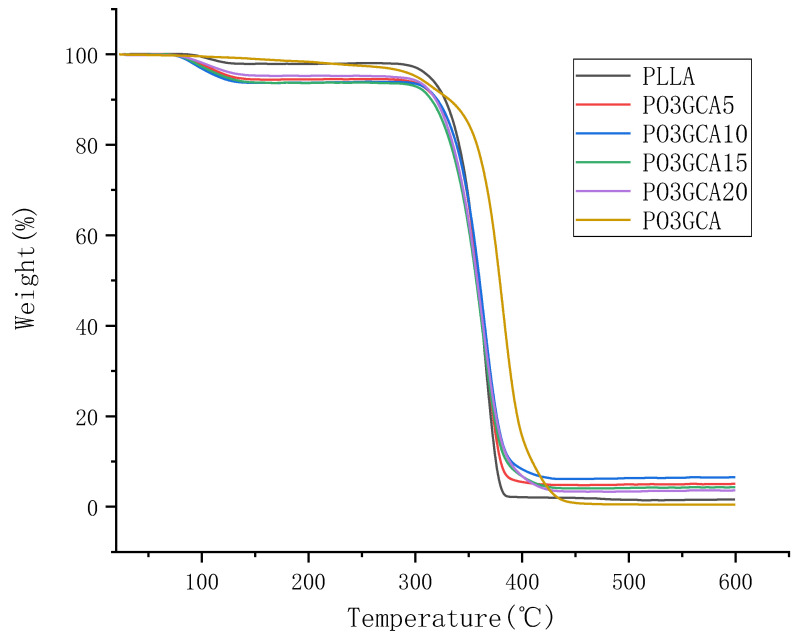
Thermogravimetric (TG) curves of polylactic-L-acid/poly(1,3-propylene glycol citrate) (PLLA/PO3GCA) films with different PO3GCA contents.

**Figure 5 polymers-15-02334-f005:**
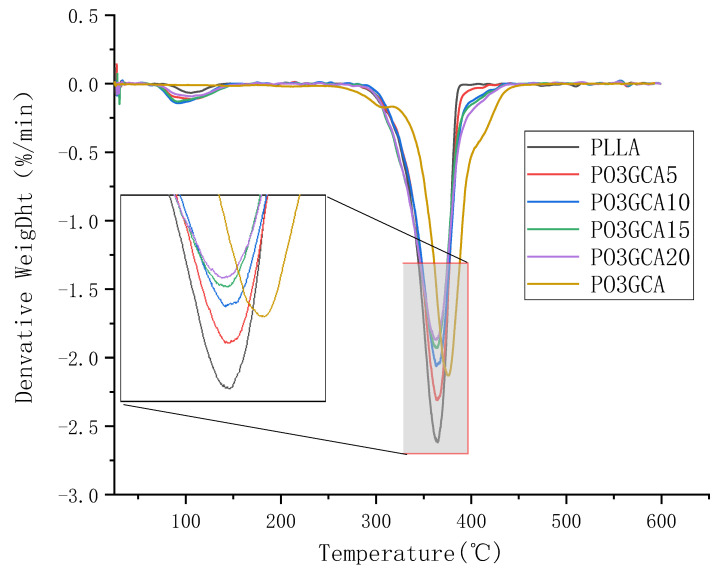
Differential thermogravimetric (DTG) curves of polylactic-L-acid/poly(1,3-propylene glycol citrate) (PLLA/PO3GCA) films with different PO3GCA contents.

**Figure 6 polymers-15-02334-f006:**
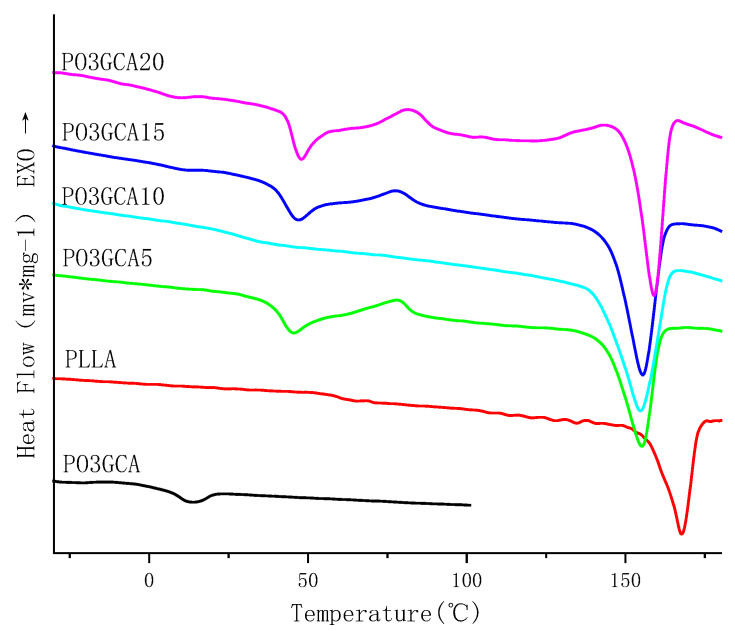
Differential scanning calorimetry (DSC) curves of polylactic-L-acid/poly(1,3-propylene glycol citrate) (PLLA/PO3GCA) films with different PO3GCA contents.

**Figure 7 polymers-15-02334-f007:**
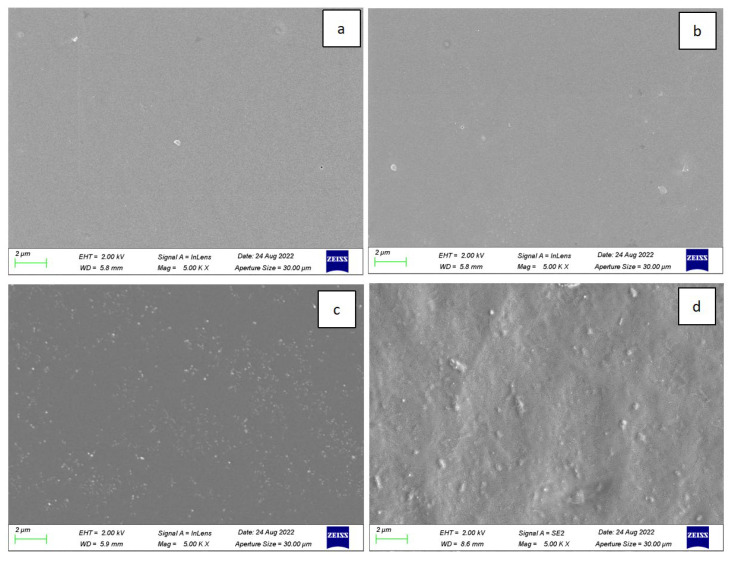
Scanning electron microscopy (SEM) images of polylactic-L-acid/poly(1,3-propylene glycol citrate) (PLLA/PO3GCA) films: (**a**) PO3GCA content, 5%; (**b**) PO3GCA content, 10%; (**c**) PO3GCA content, 15%; (**d**) PO3GCA content, 20%.

**Figure 8 polymers-15-02334-f008:**
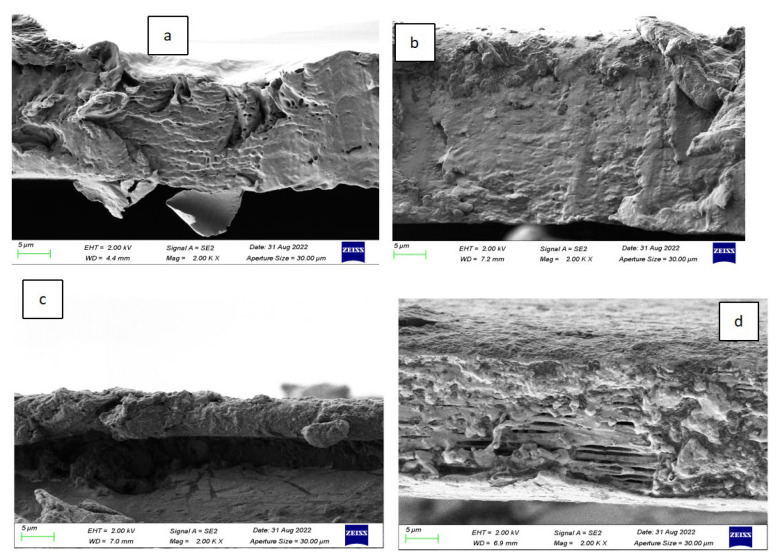
Scanning electron microscopy (SEM) images of the tensile fracture surface of polylactic acid/poly(1,3-propylene glycol citrate) (PLLA/PO3GCA) films: (**a**) PO3GCA content, 5%; (**b**) PO3GCA content, 10%; (**c**) PO3GCA content, 15%; (**d**) PO3GCA content, 20%.

**Figure 9 polymers-15-02334-f009:**
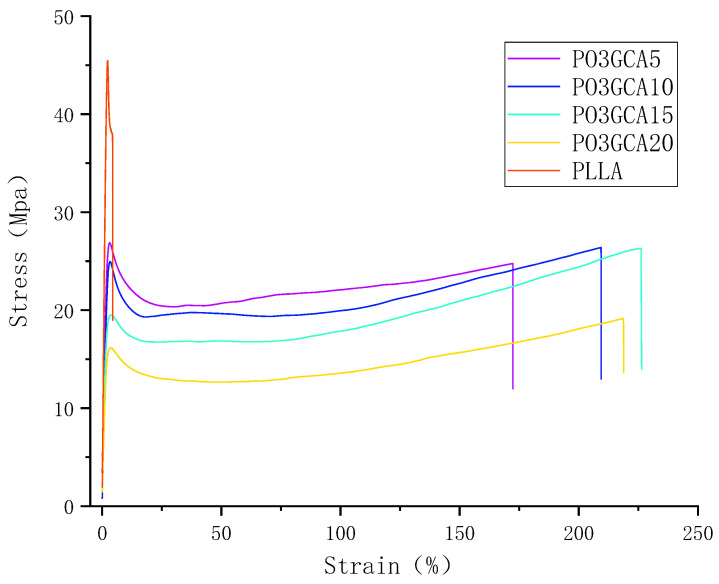
Stress–strain curves of polylactic-L-acid/poly(1,3-propylene glycol citrate) (PLLA/PO3GCA) films with different PO3GCA contents.

**Table 1 polymers-15-02334-t001:** Differential scanning calorimetry (DSC) data of PLLA thin film, PLLA/PO3GCA composite thin film, and PO3GCA.

Sample	Tg1 (°C)	Tg2 (°C)	Tcc (°C)	Tm (°C)	ΔHcc (J/g)	ΔHm (J/g)	Xc (%)
PLLA/PO3GCA5	-	41.6	78.0	155.2	5.76	27.98	24.99
PLLA/PO3GCA10	-	37.7	-	154.7	-	30.79	36.55
PLLA/PO3GCA15	8.7	43.7	78.6	155.4	3.74	28.53	31.15
PLLA/PO3GCA20	8.5	45.3	82.3	159.0	5.04	22.78	23.69
PLLA	-	62.3	123.3	165.4	35.6	39.7	4.4
PO3GCA	8.2	-	-	-	-	-	-

**Table 2 polymers-15-02334-t002:** Mechanical properties of PO3GCA/PLLA films.

PO3GCA Content (%)	Elongation AT Break (%)	Modulus of Elasticity (MPA)	Tensile Yield Stress (MPA)	Tensile Strength (MPA)	Tensile Toughness (MPA)
0	4.42	1528.00	38.12	27.28	1.47
5	172.40	1434.41	20.34	38.69	38.13
10	209.45	1339.90	19.30	35.64	44.89
15	230.03	1192.41	16.73	28.77	44.69
20	218.92	918.19	19.17	22.76	34.33

## Data Availability

The data presented in this study are available on request from the corresponding author.

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
