# Peer review of "Poly(1,3-Propylene Glycol Citrate) as a Plasticizer for Toughness Enhancement of Poly-L-Lactic Acid"

_polymers, 2023, doi:10.3390/polym15102334_

Round 1

Reviewer 1 Report

      I have carefully read this research article. They described about synthesis of PO3GCA for used as a plasticizer for PLLA. Somewhere should be improved as follow:

1. In P. 2, from “Therefore, we envisioned that PO3G or its combination with other carboxylic acids could serve as a plasticizer for PLA [15,16].”. Please check ref. nos. 15 and 16. Are they referred to this sentence?

2. In section 2.1, What are purity of CA and molybdenum trioxide in this work?

3. Scheme reaction of PO3GCA synthesis should be presented.

4. In section 2.3.1, what is %yield of PO3GCA?

5. In section 2.3.2, what is thickness of the resulting films?

6. In Fig. 2, chemical structure and peak assignment should be included.

7. From section 3.2.1.1. TGA, from “Figure 3 shows the occurrence of a weight loss from 75 °C to 140 °C for the PLA film and the PLA/PO3GCA composite films.”, what is substance to thermal decompose? It is residue solvent? It is effect on elongation at break of film samples? In section 3.3.2, what is solvent for film formation?

8. From Fig. 4, decomposition temp from DTG peaks should be reported and compared.

9. TG and DTG curves of PO3GCA should be presented for discussion and comparison.

10. In equation 2 (P. 7), this is multiplied by 100 for Xc (%)?

11. From “ωPLA denotes the percentage of PLA in the composite films”, this is weight fraction of PLA. It is not percentage of PLA.

12. In Fig. 6, are they film surfaces? The film cross-sections or film fractures from cryo-fraction should be also compared and discussed.

13. In Table 2, maximum force (N) should be removed. Tensile stress at break should be also reported and discussed. Usually, PLA has no yield point? This is effect of residue solvent from solvent evaporation?

Author Response

April 18, 2023

Dear Sir/Madam,

Thank you for reviewing our manuscript and offering valuable advice. In accordance with your suggestions, we have made the following revisions to our manuscript:

  • In P. 2, from “Therefore, we envisioned that PO3G or its combination with other carboxylic acids could serve as a plasticizer for PLA [15,16].”. Please check ref. nos. 15 and 16. Are they referred to this sentence?

Response:Thank you very much for your valuable comments.The reference location is marked wrong here, I have moved it to the correct location.

  • In section 2.1, What are purity of CA and molybdenum trioxide in this work?

Response:Thank you very much for your valuable comments.According to your suggestion,We added CA and molybdenum trioxide purity in 2.1.

  • Scheme reaction of PO3GCA synthesis should be presented.

Response:Thank you very much for your valuable comments.According to your suggestion,We added the PO3GCA synthesis roadmap.

  • In section 2.3.1, what is %yield of PO3GCA?

Response:Thank you very much for your valuable comments.According to your suggestion,We recalculated the yield of PO3GCA.

  • In section 2.3.2, what is thickness of the resulting films?

Response:Thank you very much for your valuable comments.All thin film samples prepared have a uniform thickness ranging from 0.07 to 0.08mm. We have added this data to the manuscript.

  • In Fig. 2, chemical structure and peak assignment should be included.

Response:Thank you very much for your valuable comments.According to your suggestion,We re-added the corresponding chemical structure to the spectrum and labeled the hydrogen corresponding to each peak.

  • From section 3.2.1.1. TGA, from “Figure 3 shows the occurrence of a weight loss from 75 °C to 140 °C for the PLA film and the PLA/PO3GCA composite films.”, what is substance to thermal decompose? It is residue solvent? It is effect on elongation at break of film samples? In section 3.3.2, what is solvent for film formation?

Response:Thank you very much for your valuable comments.According to your suggestion,We believe that the weight loss of this part is due to the heat evaporation of the bonded water and residual solvents in the membrane. Sorry, we did not further investigate the effects of combined water and residual solvents on the elongation at break of the material in this manuscript. We will conduct in-depth research on this in the future. The film-forming solvent used in our experiment is 1,4-dioxane.

  • From Fig. 4, decomposition temp from DTG peaks should be reported and compared.

Response:Thank you very much for your valuable comments.According to your suggestion,We rewrote this part according to your requirements, and added the TGA and DTG curves about PO3GCA.

  • TG and DTG curves of PO3GCA should be presented for discussion and comparison.

Response:Thank you very much for your valuable comments.According to your suggestion,We have supplemented the TGAH and DTG curves of PO3GCA according to your requirements and analyzed them.

  • In equation 2 (P. 7), this is multiplied by 100 for Xc (%)?

Response:Thank you very much for your valuable comments.According to your suggestion,We have perfected the formula according to your suggestion.

  • From “ωPLA denotes the percentage of PLA in the composite films”, this is weight fraction of PLA. It is not percentage of PLA.

Response:Thank you very much for your valuable comments.According to your suggestion,According to your suggestion, we have changed "PLA percentage" to "PLA quality percentage".

  • In Fig. 6, are they film surfaces? The film cross-sections or film fractures from cryo-fraction should be also compared and discussed.

Response:Thank you very much for your valuable comments.According to your suggestion,FIG. 6 is the SEM image of the surface of the film. The SEM image of the tensile fracture surface of the film is discussed in the later part.

  • In Table 2, maximum force (N) should be removed. Tensile stress at break should be also reported and discussed. Usually, PLA has no yield point? This is effect of residue solvent from solvent evaporation?

Response:Thank you very much for your valuable comments.According to your suggestion,We have removed the maximum force as per your request and replaced it with tensile strength. The appearance of the yield point of polylactic acid film may be caused by the plasticizing effect of residual solvents.

Table 2. Mechanical properties of PO3GCA/PLA films

PO3GCA CONTENT

(%)

ELONGATION AT BREAK

(%)

Modulus of elasticity

(Mpa)

TENSILE YIELD STRESS

(Mpa)

TENSILE STRENGTH(Mpa)

TENSILE TOUGHNESS

(Mpa)

0

4.42

1528.00

38.12

27.28

1.47

5

172.40

1434.41

20.34

38.69

38.13

10

209.45

1339.90

19.30

35.64

44.89

15

230.03

1192.41

16.73

28.77

44.69

20

218.92

918.19

19.17

22.76

34.33

Reviewer 2 Report

The manuscript entitled “Poly(1,3-propylene glycol citrate) as a Plasticizer for Toughness Enhancement of Poly-L-lactic Acid” reports on the esterification of commercial poly(1,3-propylene  glycol), with citric acid, its use as plasticizer for the preparation of PLLA films by means of solution casting, beside an extended characterization of the obtained films.
The subject of bio based plastics is certainly of great interest, at the present moment.
In my opinion the manuscript must be improved under both the experimental aspect, in particular concerning the DSC measurements, and the writing accuracy.

Hereby I report some suggestions.

2.1. Raw Materials
Further details on the PLA polymer would be worthy, like in: Yoo, Hyeong Min, Jeong, Su-Yeon and Choi, Sung-Woong. "Analysis of the rheological property and crystallization behavior of polylactic acid (Ingeo™ Biopolymer 4032D) at different process temperatures" e-Polymers, vol. 21, no. 1, 2021, pp. 702-709. https://doi.org/10.1515/epoly-2021-0071, where it is reported “For PLA, we used Ingeo Biopolymer 4032D in the form of pellets from NatureWorks LCC. The PLA has the ratio of l- to d-lactide of 28:1 and the melting point of 170°C. “

Is CA anhydrous or hydrated? Judging from the broad band in the OH stretching region of the iR spectrum, it looks hydrated.

2.2. Experimental equipment
Was the X-ray diffractometer (line 81) used to determine the crystallinity of the samples?

2.3.2 Preparation of PLA/PO3GCA films
State the solvent employed for the casting solutions, maybe methylene chloride.

2.4.1. 1H NMR spectroscopy characterization of PO3GCA
Was deuterated methylene chloride actually used as a solvent? No residual proton signal of the solvent is detectable in the spectrum at 5.32 ppm, see e.g. https://cil.showpad.com/share/60gq0jltDaWybfD9PsgGX.  Deuterated chloroform is the most common solvent, being much cheaper. In ref. 18 the 1H NMR spectrum of PO3G was run in CDCl3.

2.4.4. Differential scanning calorimetry (DSC) characterization of the PLA/PO3GCA films
Report the employed  cooling and heating rates.

In order to erase their thermal history, the films were heated up to a temperature lying below the melting temperature. Usually, the samples are heated above the melting temperature.
Can the author discuss in detail this point?
As a matter of fact it is important not to erase the thermal history of the samples because the preparation method, namely by solution casting, can impact on the degree of crystallinity of the sample.

I suggest the authors to reconsider the DSC measurements and to repeat according to the procedure adopted in the following recent paper :
Jianwei Guo, Xiao Liu, Ming Liu, Miaomiao Han, Yadong Liu, Shengxiang Ji, Effect of molecular weight of Poly(ethylene glycol) on plasticization of Poly(ÊŸ-lactic acid), Polymer, 223, 2021, 123720, https://doi.org/10.1016/j.polymer.2021.123720

For completeness sake, the DSC measurement might be carried out for the new plasticizer PO3GCA, as well

2.4.5 Field-emission scanning electron microscopy (SEM)
Describe the film fracture procedure

3.1 Characterization of PO3GCA
The quality of the graphics of Figure 1 is poor. Redraw it at higher resolution.
In the Figure 1 are reported enlargements of the IR spectra that were recorded on the 400-4000 wavenumber range. The indication of the position of a free main peaks could be useful , e.g. to appreciate the shift of the C=O stretching of CA upon esterification.

lines 159-167: Formatting of the chemical formulas: the numbering as subscript, whereas “1” in “1H” is a superscript.

line 162: add that the proton signals of PO3G were assigned according to the literature: ref.18 and distinguish better the different CH2 groups, (see Fig. 1 (a) of ref 18) by using an underline.

3.2.2. Mechanical properties of the PLA/POGCA films
Improve the formatting of Table 2

References

Missing author names in several reference. In many cases there is maximum of three names reported, e.g. 2, 3, 4. However, it is not an adopted general rule as in other references more names are listed, but not all e.g. ref. 5. Names instead of surnames in ref. 23

line 337, ref. 1: the year is wrong. It is 2008, not 2010

line 338, ref 2: missing page number

line 365, ref 15: missing volume number

line  371, ref 18: the year is wrong. It is 2018, not 2017

Author Response

April 18, 2023

Dear Sir/Madam,

Thank you for reviewing our manuscript and offering valuable advice. In accordance with your suggestions, we have made the following revisions to our manuscript:

The manuscript entitled “Poly(1,3-propylene glycol citrate) as a Plasticizer for Toughness Enhancement of Poly-L-lactic Acid” reports on the esterification of commercial poly(1,3-propylene  glycol), with citric acid, its use as plasticizer for the preparation of PLLA films by means of solution casting, beside an extended characterization of the obtained films.
The subject of bio based plastics is certainly of great interest, at the present moment.
In my opinion the manuscript must be improved under both the experimental aspect, in particular concerning the DSC measurements, and the writing accuracy.

Hereby I report some suggestions.

  • 1Raw Materials
    Further details on the PLA polymer would be worthy, like in: Yoo, Hyeong Min, Jeong, Su-Yeon and Choi, Sung-Woong. "Analysis of the rheological property and crystallization behavior of polylactic acid (Ingeo™ Biopolymer 4032D) at different process temperatures" e-Polymers, vol. 21, no. 1, 2021, pp. 702-709. https://doi.org/10.1515/epoly-2021-0071, where it is reported “For PLA, we used Ingeo Biopolymer 4032D in the form of pellets from NatureWorks LCC. The PLA has the ratio of l- to d-lactide of 28:1 and the melting point of 170°C.

Response:Thank you very much for your valuable comments. We have carefully read and cited the references you provided based on your suggestions. And then we explained the PLA manufacturer and product model in 2.1. The physical and chemical properties of this product are displayed on the official homepage of Nature Works, so we did not give a detailed description of it. We did use citric acid monohydrate, thank you very much for your correction, we have modified this part of the content.

  • 2. Experimental equipment
    Was the X-ray diffractometer (line 81) used to determine the crystallinity of the samples?

Response:Thank you very much for your valuable comments.According to your suggestion,We prepared to test the XRD data of the film in the early stage to obtain the crystallization data of the film, but we gave up the test due to some other reasons. We have deleted this part of the content.

  • 3.2 Preparation of PLA/PO3GCA films
    State the solvent employed for the casting solutions, maybe methylene chloride.

Response:Thank you very much for your valuable comments.According to your suggestion,The solution used to prepare the film is 1, 4-dioxane, and we have added this part in 2.3.2.

  • 4.1. 1H NMR spectroscopy characterization of PO3GCA
    Was deuterated methylene chloride actually used as a solvent? No residual proton signal of the solvent is detectable in the spectrum at 5.32 ppm, see e.g. https://cil.showpad.com/share/60gq0jltDaWybfD9PsgGX.  Deuterated chloroform is the most common solvent, being much cheaper. In ref. 18 the 1H NMR spectrum of PO3G was run in CDCl3.

Response:Thank you very much for your valuable comments.According to your suggestion,We did use deuterium chloroform, there was a typo, we have corrected that part of the content.

  • 4.4. Differential scanning calorimetry (DSC) characterization of the PLA/PO3GCA films
    Report the employed  cooling and heating rates.
    In order to erase their thermal history, the films were heated up to a temperature lying below the melting temperature. Usually, the samples are heated above the melting temperature.
    Can the author discuss in detail this point?
    As a matter of fact it is important not to erase the thermal history of the samples because the preparation method, namely by solution casting, can impact on the degree of crystallinity of the sample.
    I suggest the authors to reconsider the DSC measurements and to repeat according to the procedure adopted in the following recent paper :
    Jianwei Guo, Xiao Liu, Ming Liu, Miaomiao Han, Yadong Liu, Shengxiang Ji, Effect of molecular weight of Poly(ethylene glycol) on plasticization of Poly(ÊŸ-lactic acid), Polymer, 223, 2021, 123720, https://doi.org/10.1016/j.polymer.2021.123720
    For completeness sake, the DSC measurement might be carried out for the new plasticizer PO3GCA, as well

Response:Thank you very much for your valuable comments. 

(1)We use a heating and cooling rate of 10 ℃/min.We have added this measurement parameter to the manuscript.

(2)We have carefully read and cited the references you recommended, and we acknowledge that using DSC results that eliminate thermal history to describe our thin films is inappropriate.

(3)According to your suggestion, we conducted DSC test again for all films, and added DSC test for PO3GCA.

  • 4.5 Field-emission scanning electron microscopy (SEM)
    Describe the film fracture procedure

Response:Thank you very much for your valuable comments.According to your suggestion,The test procedure for fractured film is as follows:The fracture section of the thin film was obtained using a universal tension machine at a tensile speed of 10mm/min, and was subsequently used to conduct scanning electron microscopy (SEM) analysis of the fracture surface.

  • 1 Characterization of PO3GCA
    The quality of the graphics of Figure 1 is poor. Redraw it at higher resolution.
    In the Figure 1 are reported enlargements of the IR spectra that were recorded on the 400-4000 wavenumber range. The indication of the position of a free main peaks could be useful , e.g. to appreciate the shift of the C=O stretching of CA upon esterification.
    lines 159-167: Formatting of the chemical formulas: the numbering as subscript, whereas “1” in “1H” is a superscript.
    line 162: add that the proton signals of PO3G were assigned according to the literature: ref.18 and distinguish better the different CH2 groups, (see Fig. 1 (a) of ref 18) by using an underline.

Response:Thank you very much for your valuable comments.According to your suggestion,According to your suggestion, we have redrawn the infrared spectrum. And fixed a chemical formatting error in lines 159-167. At the same time, we labeled the proton signals of these substances according to the way of reference 18.

  • 2.2. Mechanical properties of the PLA/POGCA films
    Improve the formatting of Table 2

Response:Thank you very much for your valuable comments.According to your suggestion,We have modified the content of Table 2 according to your comments.

  • References
    Missing author names in several reference. In many cases there is maximum of three names reported, e.g. 2, 3, 4. However, it is not an adopted general rule as in other references more names are listed, but not all e.g. ref. 5. Names instead of surnames in ref. 23
    line 337, ref. 1: the year is wrong. It is 2008, not 2010
    line 338, ref 2: missing page number
    line 365, ref 15: missing volume number
    line  371, ref 18: the year is wrong. It is 2018, not 2017

Response:Thank you very much for your valuable comments.According to your suggestion,We have modified the format of references 1,2,15 and 18 according to your requirements.

  1. Fukushima, K.;Kimura, Y. An efficient solid-state polycondensation method for synthesizing stereocomplexed poly(lactic acid)s with high molecular weight. Journal of Polymer Science Part A Polymer Chemistry, 2008, 46(11), 3714-3722.
  2. X Montané .; Montornes J M .; Nogalska A .; M. Olkiewicz .; B Tylkowski . Synthesis and synthetic mechanism of Polylactic acid. Physical Sciences Reviews,2020, 5, 12-12.
  3. Maiza M , Benaniba M T .; Massardier-Nageotte, V. Plasticizing effects of citrate esters on properties of poly(lactic acid). Journal of Polymer Engineering, 2016,36, 371-380
  4. Cong Z.; Lua-Cheng H.; Gui-You W. A novel thermosensitive triblock copolymer from 100% renewably sourced poly(trimethylene ether) glycol.Journal of Applied Polymer Science2018135, 46112.

Round 2

Reviewer 1 Report

      I have carefully read this revised article. They described about synthesis of PO3GCA for used as a plasticizer for PLLA. Somewhere should be improved as follow:

1) In P. 1, from “One of the major drawbacks of PLA is its low glass transition temperature (Tg) of 55 °C–65 °C”, it is not “low glass transition temperature”. This must change to “high low glass transition temperature”.

2) From Fig. 1, PO3G reacted only one site of CA? I think that PO3G can react on three -OH groups of CA? Reaction should be corrected including Fig. 3.

3) From section 2.3.2, 1,4-dioxane was used as a solvent for film forming. Its boiling point is 101 C. It is very difficult to evaporation. Why are the authors used 1,4-dioxane? Moreover, from “were naturally dried at room temperature for 48 h”, is it complete evaporate?

4) All Figures should be placed after said about it.

5) Equation (2) for Xc calculation should be addressed in section 2.4.4.

6) In Table 2, “MPA” must change to “MPa”.

Author Response

April21, 2023

Dear Sir/Madam,

Thank you for reviewing our manuscript and offering valuable advice. In accordance with your suggestions, we have made the following revisions to our manuscript:

  • In P. 1, from “One of the major drawbacks of PLA is its low glass transition temperature (Tg) of 55 °C–65 °C”, it is not “low glass transition temperature”. This must change to “high low glass transition temperature”.

Response:Thank you very much for your valuable comments.We have corrected the mistakes in this part according to your requirements. Thank you again for pointing out the mistakes in this part.

  • From Fig. 1, PO3G reacted only one site of CA? I think that PO3G can react on three -OH groups of CA? Reaction should be corrected including Fig. 3.

Response:Thank you very much for your valuable comments. We have corrected the error in Figure 1. After consulting literature [1], it has been found that when large groups exist near the reaction point of the esterification reaction, the esterification reaction rate will significantly decrease. Therefore, in citric acid, the esterification activity of the hydroxyl group at the 1,5 position is much higher than that of the hydroxyl group on the quaternary carbon. Therefore, we present the main reaction of citric acid esterification in Figure 1.

[1]Morrison R T , Boyd R N . Organic chemistry[M]. Allyn and Bacon, 1983.

  • From section 2.3.2, 1,4-dioxane was used as a solvent for film forming. Its boiling point is 101 C. It is very difficult to evaporation. Why are the authors used 1,4-dioxane? Moreover, from “were naturally dried at room temperature for 48 h”, is it complete evaporate?

Response:This is a very good question. In the initial stage of our research, we did use the volatile dichloromethane as the solvent for the experiment. However, during the casting process, we found that the rapid evaporation of solvents can cause phase separation between plasticizers and PLA in the film. Therefore, we chose 1,4-dioxane, which has a slower evaporation rate, as the film-forming agent and found that using 1,4-dioxane as a solvent can make the plasticizer uniformly dispersed in PLA without phase separation. However, due to the difficulty in volatilizing the 1,4-dioxane ring, a small amount of solvent remains in the membrane.

  • All Figures should be placed after said about it.

Response:Thank you very much for your valuable comments.We have moved all the data behind the corresponding text according to your requirements. Thank you again for your comments.

  • Equation (2) for Xc calculation should be addressed in section 2.4.4.

Response:Thank you very much for your valuable comments.We have moved the formula to 2.4.4 according to your request. Thank you again for your comments.

  • In Table 2, “MPA” must change to “MPa”.

Response:Thank you very much for your valuable comments.We have revised this part according to your requirements. Thanks again for your advice.

Reviewer 2 Report

The revised version of the manuscript entitled “Poly(1,3-propylene glycol citrate) as a Plasticizer for Toughness Enhancement of Poly-L-lactic Acid” is clearer and more precise than the original one. Actually, the authors rewrote some parts in a more extended form, added information required for reproducibility sake and corrected some small mistakes.

Author Response

April21, 2023
Dear Sir/Madam,
Thank you for reviewing our manuscript and offering valuable advice. In accordance with your suggestions, We carefully checked and corrected the grammar of the manuscript.